# The Immunogenicity and Safety of *Mycobacterium tuberculosis*-*mosR*-Based Double Deletion Strain in Mice

**DOI:** 10.3390/microorganisms11082105

**Published:** 2023-08-18

**Authors:** Rachel E. Hildebrand, Chungyi Hansen, Brock Kingstad-Bakke, Chia-Wei Wu, Marulasiddappa Suresh, Adel Talaat

**Affiliations:** 1Department of Pathobiological Sciences, University of Wisconsin-Madison, 1656 Linden Drive, Madison, WI 53706, USA; rhildebrand@wisc.edu (R.E.H.); chungyi.hansen@wisc.edu (C.H.); babakke@wisc.edu (B.K.-B.); chiaweiwu@wisc.edu (C.-W.W.); suresh.marulasiddappa@wisc.edu (M.S.); 2Vireo Vaccine International, Middleton, WI 53562, USA

**Keywords:** tuberculosis vaccine, mycobacteria, safety, immunology

## Abstract

*Mycobacterium tuberculosis* (*M. tuberculosis*) remains a significant global health threat, accounting for ~1.7 million deaths annually. The efficacy of the current vaccine, *M. bovis* BCG, ranges from 0 to 80% in children and does not prevent adulthood tuberculosis. We explored the immune profile and safety of a live-attenuated *M. tuberculosis* construct with double deletions of the *mosR* and *echA7* genes, where previously, single mutations were protective against an *M. tuberculosis* aerosol challenge. Over 32 weeks post-vaccination (WPV), immunized mice with *M. tuberculosis*Δ*mosR*Δ*echA7* (double mutant) were sacrificed to evaluate the vaccine persistence, histopathology, and immune responses. Interestingly, despite similar tissue colonization between the vaccine double mutant and wild-type *M. tuberculosis*, the vaccine construct showed a greater reaction to the ESAT-6, TB.10, and Ag85B antigens with peptide stimulation. Additionally, there was a greater number of antigen-specific CD4 T cells in the vaccine group, accompanied by significant polyfunctional T-cell responses not observed in the other groups. Histologically, mild but widely distributed inflammatory responses were recorded in the livers and lungs of the immunized animals at early timepoints, which turned into organized inflammatory foci via 32WPV, a pathology not observed in BCG-immunized mice. A lower double-mutant dose resulted in significantly less tissue colonization and less tissue inflammation. Overall, the double-mutant vaccine elicited robust immune responses dominated by antigen-specific CD4 T cells, but also triggered tissue damage and vaccine persistence. The findings highlight key features associated with the immunogenicity and safety of the examined vaccine construct that can benefit the future evaluation of other live vaccines.

## 1. Introduction

*Mycobacterium tuberculosis*, the causative agent of tuberculosis, currently infects an estimated quarter of the world’s population, resulting in ~1.7 million deaths annually [1]. The only approved vaccine for *M. tuberculosis* is the live-attenuated vaccine *M. bovis* Bacillus Calmette–Guerin (BCG), the most administered vaccine worldwide. While BCG protects children from disseminated tuberculosis, it has a limited efficacy in adults [2,3,4]. There are many explanations for BCG’s lack of efficacy, such as little central memory cell development in the lungs [5] and being too attenuated to produce robust immune responses [6], as well as numerous genetic and in vivo transcriptional differences between *M. tuberculosis* and BCG that could make it harder for BCG to induce protection against *M. tuberculosis* [7,8]. The continuing high disease burden of *M. tuberculosis*, as well as BCG’s failures as a vaccine, highlight the need for continuing vaccine discovery to better control tuberculosis and improve tuberculosis outcomes worldwide. A historical and current strategy for *M. tuberculosis* vaccine development has been using live-attenuated vaccines (LAVs) made by modifying strains of BCG [9,10], and even using *M. tuberculosis* as the backbone [11,12,13]. LAVs with an *M. tuberculosis* backbone have antigens that BCG lacks and potentially provide a more robust immune response. In fact, another LAV candidate known as MTBVAC, *Mycobacterium tuberculosisΔphoPΔfadD26,* is in clinical trials and yielding promising results [13,14,15]. According to the guidelines established by the WHO (Geneva conference I and II), attenuated vaccines with an *M. tuberculosis* backbone must have at least two gene deletions for ensured safety [16]. Based on in vivo murine transcriptional studies of *M. tuberculosis*, we selected several candidates to test as live-attenuated vaccines in previous studies [17,18].

From these studies, we chose *mosR* as a deletion candidate, as it was the most highly upregulated gene in murine lungs at multiple timepoints (28, 45, and 60 days post-infection) when compared to *M. tuberculosis* in culture; upon further study, it was revealed to be an operon controlling gene involved in late-stage mycobacterial survival [17,19]. The single deletion of *mosR* in the H37Rv backbone was shown to be highly attenuated via the aerosol route and the single deletion in the CDC1551 backbone was shown to be a safe and effective vaccine in a previous study [19,20]. In another transcriptional study, a group of 32 consecutive genes were found to be highly expressed in mouse lungs, of which *echA7* was one [18]. This group of genes was named iVEGI (in vivo-expressed genomic island) and most operons within iVEGI have been shown to be required for full virulence within animal hosts [19,21]. An echA7 single deletion mutant was tested and showed a log reduction in colonization, as well as 100% survival in a study lasting 46 weeks as compared to H37Rv-infected mice, which survived, on average, for 31 weeks post-challenge [21]. These single deletions were both tested separately as LAV candidate strains, *M. tuberculosis* H37RvΔ*echA7* and *M. tuberculosis* CDC1551Δ*mosR*, and characterized in comparison to BCG vaccination using aerosol vaccine/challenge trials [20]. Both LAVs offered superior protection at 30 and 60 days post-challenge and had similar or better T-cell responses compared to BCG. In this report, we will further examine the safety and immunogenicity of an *M. tuberculosis* construct with deletions of both the *echA7* and *mosR* genes.

A significant hurdle against the discovery of new tuberculosis vaccines is the lack of knowledge on the correlates of immunity that drive protection against an *M. tuberculosis* challenge [22]. Correlates of immunity are important for the validation of experimental vaccines, the determination of most the promising candidates, and can inform key clinical questions such as those on dosage, frequency, and immune durability. Although the LAV candidates of *M. tuberculosis* H37RvΔ*echA7* and *M. tuberculosis* CDC1551Δ*mosR* both resulted in protection [20], the identified correlates of this protection were not well-characterized and the immunogenicity of the construct may change with the inclusion of a second gene deletion. Previously, a vaccine candidate known as MVA85A completed an efficacy trail, where it induced higher levels of antigen-specific T-cell responses than BCG in mice, but did not offer any increased protection in other models, despite those responses previously being identified as indicators of a protective response [22,23,24]. Such results highlight the need to identify reliable measures for determining vaccine efficacy and that more thorough immune characterizations of protective vaccine candidates are needed.

Here, we created a mutant with deleted *echA7* and *mosR* genes in *M. tuberculosis* CDC1551 to meet the WHO standards for LAVs in the backbone of CDC1551, known here as the double-mutant vaccine. We characterized this mutant safety and immunogenicity when given via subcutaneous injection in comparison to BCG and the wild-type (WT) parental strain, *M. tuberculosis* CDC1551. We determined that the double-mutant vaccine was able to persist in vivo and caused moderate inflammation in the tissues over the 32-week study, though without consequences on weight or mortality. The double-mutant vaccine also produced superior IFNγ, TNFα, IL-17, and IL-2 CD4 and CD8 T-cell responses compared to BCG and even the WT CDC1551. Interestingly, the surface cell markers revealed little difference between the *M. tuberculosis*-specific CD4 T-cells of the vaccinated mice in terms of different cell populations, but offered interesting comparisons between the two antigens tested, Ag85B and ESAT-6. Finally, we examined the consequences of dosage differences on the vaccine persistence, showing that a log reduction in the vaccine dose resulted in a 75% reduction in the colonization levels and associated tissue lesions.

## 2. Materials and Methods

### 2.1. Bacterial Strains and Growth Media

*M. bovis* BCG (Pasteur TMC 1011) and *M. tuberculosis* CDC1551 were originally received from the American Type Culture Collection (ATCC) and maintained in Talaat’s laboratory. The double-mutant (*M. tuberculosis* CDC1551ΔmosRΔechA7) vaccine was constructed in the lab, utilizing a previously constructed strain (*M. tuberculosis* CDC1551ΔmosR) that had been characterized previously [20]. Cultures for *M. tuberculosis* CDC1551, *M. tuberculosis* CDC1551ΔmosRΔechA7, and *M. bovis* BCG (BCG) were grown to ~1 OD in Middlebrook 7H9 (Remel^TM^, Lenexa, KS, USA), supplemented with 10% ADC, 0.05% TWEEN 80, hygromycin, and kanamycin when appropriate. Stocks were then washed twice, frozen in storage media (10% glycerol, 0.05% TWEEN 80, and 0.85% NaCl in deionized water), and stored at −80 °C. The dosage of the stocks was determined via plating dilutions on Middlebrook 7H10 (Remel^TM^), supplemented with ADC and incubated at 37 °C for 4–6 weeks. The mycobacterial loads in the tissues were determined by homogenizing pre-weighed tissue samples in PBS and plating them on supplemented Middlebrook 7H10, as performed previously, and the plates were supplemented with kanamycin (30 ug/mL) and hygromycin (30 ug/mL) as appropriate.

### 2.2. Phage Construction and Mycobacterial Transduction for Double-Knockout Mutant

An *M. tuberculosis* knockout mutant, Δ*mosR*, was previously constructed via specialized transduction with recombinant mycobacteriophage carrying *mosR* allelic exchange substrates into wild-type *M. tuberculosis* CDC1551 [19,20]. The allelic exchange events resulted in the disruption of the *mosR* gene and hygromycin-resistant phenotype. In order to generate a double-knockout mutant, a plasmid for *echa7* was made by amplifying kanamycin and sacB gene fragments individually into pYUB870. The backbone vector (pYUB854) was then digested with the same enzymes and ligated with purified insert fragments into a new plasmid known as pCW1702. Using this plasmid, a specialized mycobacteriophage was constructed after the *echA7* allelic exchange substrates were cloned. This mycobacteriophage was then used in specialized transduction, as performed previously [19,20], resulting in a double-knockout mutant strain selected for by the Hyg/Kan phenotype and confirmed via PCR amplification and a whole-genome sequencing analysis.

### 2.3. Mouse Vaccinations and Infections

All the animal experiments were first approved by the Institutional Animal Care and Use Committee (IACUC), University of Wisconsin–Madison. The safety and immunogenicity of the vaccines were evaluated in female C57BL/6 mice, 6–8 weeks old, obtained from Jackson Labs. The mice were housed under animal biosafety level-3 (ABSL3) conditions, according IACUC regulations. Vaccine stocks of *M. bovis* BCG Pasteur 1101, *M. tuberculosis* CDC1551, and *M. tuberculosis*Δ*mosR*Δ*echA7* were prepared by diluting frozen stocks in PBS to a dose of 10^6^ CFU/100 uL or 10^5^ CFU/100 uL and were administered subcutaneously. Sterile PBS was administered to unvaccinated groups as a control. The mice were then sacrificed at 4, 8, 16, and 32 weeks post-vaccination (WPV), with groups of N = 3–5 mice per vaccine group. The mice were monitored regularly for adverse reactions to the vaccination and their overall health, including weekly weighing. Sections of the livers, lungs, and spleens of the animals were harvested for colony count and histology, as performed previously [20].

### 2.4. Flow Cytometric Assessment of M. tuberculosis-Specific T-Cell Types and Responses

Representative numbers of the immunized mice (N = 3) from each group were euthanized and used for a flow cytometric analysis. Singe-cell suspensions from the lungs and spleens were placed in 1% Fetal Bovine Serum (GeminiBio, West Sacramento, CA, USA) RPMI media (Corning, Tewksbury, MA, USA) with 1 mg/mL of collagenase B (Roche, Palo Alto, CA, USA) and incubated at 37 °C for 45 min, with periodic shaking. The lungs and spleens were then pushed through a 70 mm cell strainer (Falcon, Wilmington DE, USA) and then the red blood cells were lysed using the 1× BD Biosciences BD Pharm Lyse^TM^ to prepare for the single-cell suspensions. For evaluating the intracellular cytokine staining (ICCS), 10^6^ cells were simulated with peptide pools of Ag85B, ESAT-6, or TB10.4 (2 ng/mL, BEI resources, NR-34828, NR-50711, and NR-34826) for 5 h at 37 °C with Brefeldin A (1 uL/mL, GolgiPlug, BD Biosciences, San Jose, CA, USA) and IL-2. Following this stimulation, the cells were stained with Dye eFluor 780 (eBiosciences, San Diego, CA, USA) for viability. The cells were then stained with fluorophore-conjugated antibodies, detecting CD4 (BUV 496, GK1.5), CD8a (BUV395, 53-6.7) surface antigens. The cells were then processed with the Cytofix/Cytoperm kit (BD Biosciences, San Jose, CA, USA). Subsequently, the cells were stained with antibodies for an intracellular antigen detection of IFNγ (APC, XMG1.2), TNFα (BV421, MP6-XT22), IL-2 (PE-CF594, JES6-5H4), and IL-17 (FITC, TC11-18H10.1). Samples were then acquired on an LSR Fortessa (BD Biosciences, San Jose, CA, USA) flow cytometer and the data were analyzed with the FlowJo software ver10.6.1 (TreeStar, Woodburn, OR, USA). The results are expressed as the percentage of CD4^+^ or CD8^+^ expression for both the stimulated cells and their paired unstimulated samples.

For the surface stain panel, cells were prepared in the same way as those for the ICCS panel and plated with 10^6^ cells per well. The cells were first stained for viability (eFluor 780) for 30 min at 4 °C. Then, MHC II tetramers from the NIH Tetramer Core Facility (Emory University, Atlanta, Georgia, USA) for Ag85B (PE) and ESAT-6 (APC) were diluted to 1:50 and added to the cells, which were then incubated at 37 °C for 90 min. Following the tetramer staining, the cells were stained with antibodies diluted in brilliant violet stain buffer (BD Horizon^TM^, San Jose, CA, USA) and incubated on ice for 30 min. The antibodies were as follows: CD127 (PerCP-Cy5.5, A7R34), CD4 (BUV496, GK1.5), CD8 (BUV395, 53-6.7), CD49a (BV605, Ha31/8), CD69 (PE-Cy7, H1.2F3), CD103 (FITC, 2E7), CD44 (BV510, IM7), CXCR3 (BV650, CXCR3-173), CD62L (PE-CF594, AF700), CD25 (APCR-700, PC61), CXCR1 (BV785, SAO11F11), and KLRG1 (BV711, 2FI). Samples were then fixed, acquired, and analyzed similar to ICCS. The results are expressed as percentages of CD4^+^ T-cells or as percentages of tetramer-positive CD4^+^ T-cells. All the antibodies for the cytokine staining were ordered from either BD biosciences, Ebiosciences, or Biolegend (San Diego, CA, USA). The gating strategy for all the flow cytometry analyses is displayed in Appendix A.

### 2.5. RNA Extraction and Quantitative Real-Time PCR

RNA was extracted from the *M. tuberculosis* cultures collected at ~1.0 OD. Samples were extracted using the Zymo Direct-zol RNA microprep Kit (Zymo Research, CA, USA), according to the manufacturer’s instructions. Using the Invitrogen SuperScript III first-strand synthesis system, cDNA synthesis was performed with 0.5 uL (50 g/uL) of random hexamers, 0.5 uL of 10 mM deoxynucleotides triphosphate, and 8 uL of RNA diluted in a total volume of 15 uL of nuclease-free water, which was then heated at 65 °C for 5 min and chilled on ice. After heating, 1 uL of 10xRT buffer, 1 uL of 0.1 M dithiothreitol, 2 uL of 25 mM MgCl_2_, 0.5 uL of RNAse-OUT, and 0.5 uL of SuperScript III enzyme were then added. The reaction was then placed in a thermocycler at: 25 °C for 5 min, 50 °C for 60 min, and 70 °C for 15 min. PCR was then performed with the following reaction mixture: 2 uL of cDNA diluted in a total of 8 uL of nuclease-free water, 10 uL of GoTaq quantitative PCR (qPCR) master mix (Promega, Madison, WI, USA), and 1 uL of each forward and reverse primers. The PCR was performed using the StepOnePlus real-time PCR system (Applied Biosystems, Foster City, CA, USA) with the following settings: 1 cycle at 95 °C for 2 min, 40 cycles at 95° for 3 s, and then 60° for 30 s. Samples were prepared in both biological and technical triplicates for each culture and the primers were *echA7*-specific or for an internal control (16S rDNA gene). Serial 5-fold dilutions of cDNA were used to establish the standard curves and temperature melt curves prior to the PCR being performed.

## 3. Results

Relationship between *mosR* deletion and *echA7* expression.

To test whether an *echA7* gene would make a good target for a second mutation in an LAV, we extracted the total RNA from the *M. tuberculosis* Δ*mosR* cultures from different genetic backgrounds (H37Rv and CDC1551) to examine the changes in the *echA7* expression. Especially in the CDC1551 strain*,* the transcripts for *echA7* were upregulated in the Δ*mosR* strains, indicating that it is under the negative control of the global transcriptional regulator *mosR* (Figure 1A); hence, *echA7* deletion could further reduce the mutant virulence. For the double gene deletions, we selected the CDC1551 backbone, a relatively recent clinical isolate of *M. tuberculosis* that maintains its virulence [25] and could yield a better vaccine immunogenicity than the H37Rv laboratory strain. Using mycobacteriophages [26], an *echA7* allelic exchange substrate, the kanamycin cassette and *sacB* cassette were deleted/inserted using specialized transduction (Figure 1B). The resulting mutant (Figure 1C) was selected on media supplemented with hygromycin and kanamycin and confirmed using PCR amplification and whole-genome sequencing.

*M. tuberculosis*Δ*mosR*Δ*echA7* is able to persist in mice.

To evaluate the safety of the live-attenuated vaccine (LAV) candidate, *M. tuberculosis* Δ*mosR*Δ*echA7* (double mutant) C57BL6 mice were immunized subcutaneously and weighed weekly. The majority of both the lungs and spleens of the double-mutant vaccine mice were colonized throughout the study, containing similar levels of bacterial persistence as that of the parental CDC1551 strain (Figure 2). Interestingly, the number of tissues colonized for the LAV dropped from 100% colonization at 4 WPV to 40% at 8 WPV, before increasing back to 100% at 16 and 32 WPV. As expected, no BCG colonies were isolated from the lungs, while colonies were sporadically found in the spleens. For the histopathology, both the vaccine mutant and CDC1551 caused mild/moderate inflammation in the lungs, with a large variation within each group at 4, 8, and 16 WPV (Figure 3). At 32 WPV, inflammation became more severe in the double-mutant vaccine group, having a greater percentage of tissue affected. Additionally, more prominent histiocytic foci began forming, evidencing a more progressive infection in the double-mutant vaccine group. The BCG-vaccinated mice had no increased inflammation compared to PBS throughout the study (Figure 3). Additionally, no mortality due to the vaccination nor differences in weight occurred between the groups throughout the 32 WPV period (Appendix A). Interestingly, the double-mutant vaccine showed inflammation in the liver at 4 and 8 WPV, which was then resolved by 16 WPV, unlike that in other organs. Overall, both the CDC1551 and double mutant induced tissue damage and were able to persist in the mice, without causing any mortality or notable morbidities. This mortality is striking in comparison to aerosol infection, where CDC1551-infection-induced mortality begins at 16 weeks post-infection (WPI), with the LD50 occurring around 32 to 36 (WPI) [27,28]. The BCG mice showed minimal to no tissue damage and no inflammation, with low levels of persistence underscoring its safety, as expected.

The immune profile of the double mutant vaccine.

To characterize the immune responses developed against the vaccination, lung and spleen single-cell suspensions were isolated from the 4-, 8-, 16-, and 32-WPV vaccinated mice of all groups. The cell suspensions were stained with MHC II tetramers specific for Ag85B and ESAT-6, which attach onto T-cell receptors that match those antigens, as well as anti-CD4^+^, anti-CD8a^+^, and various other surface cell markers for distinguishing cell types. The T-cells were characterized as memory effector (CD69^LO^ and CD62^LO^), central memory (CD69^LO^ and CD62^HI^), resident memory (CD103^HI^, CD69^HI^, and CD62^LO^), and T-regulatory (CD25^HI^ and CD127^LO^) in the CD4^+^ CD44^+^ cells responding to Ag85B or ESAT-6. As BCG does not contain ESAT-6, this was used as a second negative control for those groups. Interestingly, the mutant vaccine produced statistically higher Ag85B-responding cells than both CDC1551 and BCG at 4 WPV and 32 WPV (Figure 4). For ESAT-6, statistically higher responding populations were identified at the 8 WPV and 32 WPV timepoints from BCG (negative control), although these were still higher than the CDC1551-vaccinated mice (Figure 4). Although variable numbers of antigen-responding groups were identified, the T-cell populations of effector memory, resident memory, T-reg, and central memory remained very similar between the groups (Figure 4). While no large differences occurred between the vaccinated groups, there were differences between the types of cells responding to the two Ag85B and ESAT-6 antigens. Specifically, T-reg populations were largely undetected in ESAT-6, while they accounted for the second-largest population behind effector memory for Ag85B. Resident memory cells were mostly ESAT-6-responding and only Ag85B had a significant population of central memory cells at any point. Additionally, larger numbers of ESAT-6 cells were apparent at all timepoints, except for at 32 WPV. In summary, the double-mutant vaccine was able to produce more T-cells that responded to *M. tuberculosis*-specific antigens than the other control groups. However, there were no apparent differences in what types of T-cells these responding cells were, perhaps partially due to the low counts of responders in the controls.

The double-mutant vaccine produces superior CD4^+^ and CD8^+^ T-cell cytokine responses.

In addition to the surface characterization, we also sought to characterize the cytokine production upon stimulation with the key *M. tuberculosis* antigens of ESAT-6, Ag85B (CD4s), and TB10.4 (CD8s). Single-cell suspensions were stimulated with peptide pools or left unstimulated and then stained for CD4, CD8, IL-17, IFNγ, TNFα, IL-17, and IL-2 (Figure 5A–D). Notably, the double-vaccine mutant elicited the highest cytokine responses across the stimulation pools, frequently higher than that of the CDC1551 group as well. At 16 WPV, the cytokine levels were diminished for the ESAT-6 and Ag85B stimulants compared to the earlier timepoint of 8 WPV. For Ag85B, this decreased response carried on into the 32 WPV timepoint as well. Despite being the only available tuberculosis vaccine, BCG had very few positive responses towards Ag85B and TB10.4 stimulation for all cytokines. Finally, there were several cytokines for CDC1551 at which, upon stimulation, decreased. Most notably, the IL-2 and IL-17 levels in the stimulated cells were less than the unstimulated levels at 32 WPV for the CDC WT group. Overall, the double-mutant vaccine was more immunogenic and reactive to the various stimulations than the other vaccination groups, although it still exhibited a decrease in reactivity at 16 WPV.

Persistence of a low dose of double-mutant vaccine.

To evaluate the impact of a lower dose on the vaccine safety, the mouse groups were vaccinated with a target dose of 10^6^ CFU/100 uL or lower dose of 10^5^ CFU/100 uL. At 16 WPV, the mice were sacrificed, and their lungs, spleens, and livers were taken for bacterial enumeration and histopathology. The mice vaccinated with a lower dose had only one spleen from one animal with detectable colonization levels (~30 CFU/g), while the animals with the higher dose had 100% colonization in their spleens and 75% colonization in their lungs (Figure 6).

## 4. Discussion

Developing effective vaccines against *M. tuberculosis* remains critical for tuberculosis control and is a global health priority. Even with a ~90% BCG vaccination rate in high-burden areas, the annual death rate for TB remains around 1.7 million [29]. Live-attenuated vaccines continue to be a strong area of potential development, with many recent live-attenuated vaccines utilizing knockouts or insertion into the current BCG vaccine [30]. Vaccine constructs such as BCGΔBCG1419 utilize an additional gene knockout and result in higher polyfunctional T-cells and reduced inflammation [31]. Other vaccines such as BCG::*phoPR*, BCG::RD1, and BCG::ESAT6-PE25SS utilize gene insertion to enhance BCG immunogenicity, although this also results in increased tissue inflammation as well [32,33,34]. Another alternative includes both an insertion and deletion designed to enhance BCG’s apoptotic properties, known as BCGΔ*ureC*::hlyΔ*nuoG* [35,36]. However, newer, effective vaccines, such as MTBVAC, utilize an *M. tuberculosis* background containing *fadD26* and *phoP* gene deletions [13,14,15]. MTBVAC was similarly effective compared to BCG against an ultra-low dose of *M. tuberculosis* in the macaque model [37], which allowed this vaccine to progress to clinical trials. Even with MTBVAC’s success, new vaccine candidates with additional knockouts in MTBVAC are also being developed and tested [38,39], highlighting the demand for and necessity of continuing to explore options for a better TB vaccine.

Understanding the need for novel vaccine candidates, our lab developed CDCΔ*mosR* and H37RvΔ*echA7* and, upon further study, found that they made for promising vaccine candidates [20]. After discovering that CDCΔ*mosR* upregulated *echA7* in comparison to its wild-type parental strain, we made a double-deletion mutant as a vehicle for better understanding the in vivo dynamics of live-attenuated vaccine candidates and their potential. We found that our double-mutant vaccine was able to persist at low levels in the lungs and spleens of aerosol-infected animals and showed, in fact, higher colonization levels than the CDC1551 parent strain when given via subcutaneous injection. While most live-attenuated vaccines for tuberculosis have not been evaluated for prolonged periods, where the double vaccine mutant had the highest bacterial burdens and worst lesions at 32 WPV, the other live-attenuated vaccine candidates had similar levels of persistence. This has occurred even in the far more attenuated backbone of BCG. For example, the BCG::RD1 and BCG::pYUB vaccine candidates delivered intravenously had comparable or higher levels of persistence as the double vaccine mutant at 8 WPV [33,34]. Additionally, MTBVAC (dosage of 5 × 10^5^ CFU) in BALB/c mice showed similar levels of persistence at 4 WPV and 8 WPV as our vaccine mutant (dosage of 10^6^ CFU), although at later timepoints, MTBVAC decreased in burden, while the double mutant vaccine increased in this study [39,40]. However, when comparing this to our dose experiment, the mice given only a 10^5^ CFU dose of the double-mutant vaccine had similar levels of persistence at 16 WPV as MTBVAC [39,40]. Given these results, our study highlights both the need to evaluate the dose as well as longevity of safety studies on live-attenuated vaccines.

The vaccine mutant was able to cause significant pathological damage in vivo, although notably without mortality or morbidity (weight loss) throughout the 32-week study. However, inflammation at the earlier timepoints of 4 WPV and 8 WPV did not vary greatly in some other LAV vaccines being tested [34]. Interestingly, while the lung had increased inflammation and pathological damage over time, the liver exhibited the most severe inflammation at 8 WPV, which then resolved, with very little inflammation being seen by 32 WPV for the double-mutant vaccine. As expected, the BCG groups exhibited little to no inflammation in any tissue. Our initial bacterial and pathological findings for the double-mutant vaccine seemed contrary to our prior results, given that the single knockouts of the same genes were unable to persist and caused minimal damage at similar doses. However, a lower-dose vaccine demonstrated that a 1-log reduction in the vaccine dosage reduced the colonized mice by 75% in the spleen and 100% in the lungs (Figure 6). Even in the 10^6^ CFU dosage control for the dosage experiment, the lung bacterial loads were a log lower than those in the original safety study. This additional dosage study demonstrated how just a small difference in dosing can affect the colonization results. This also may account for the difference between the CDC WT and double gene deletion as well. Interestingly, the CDC WT colonization in the lung dropped to only 20% at 32 WPV, suggesting that even low doses of WT strains via the subcutaneous route can be controlled by the host’s immune system, underlying the potential of LAV vaccines. However, the ability to persist long-term in the lung, in addition to causing organized lesions, shown at 32 WPV (Figure 3) for the double-mutant vaccine is worrisome for safety considerations. Finally, although BCG’s low inflammation evident in histopathological sections and low colonization levels are a testament to its safety, its inability to reach the lung may be the reason for its limited ability to induce significant protection or detectable immune responses there.

Live-attenuated vaccine development balances safety with immunogenicity. Some recombinant BCG vaccines, when delivered via other routes, show an increased pathogenicity, but these delivery methods also result in the best immune responses [34]. The double-mutant vaccine produced robust immune responses after stimulation with all peptide pools at the 8 WPV and 32 WPV timepoints, even in comparison to the CDC WT vaccination. There were also populations of polyfunctional T-cells, in particular the IFNγ^+^ TNFα^+^ IL-2^+^ populations that have previously been noted as correlates for immunity in mice [41,42]. Additionally, TB10.4 stimulation produced CD8 T-cell responses of INFγ and TNFα at both 16 and 32 WPV for the double-mutant vaccine group. Notably, all the immune response reactions to stimulation were reduced at 16 WPV for the *M. tuberculosis* backbone-immunized groups. A possible explanation for this is that active TB infection can cause reduced cytokine levels as T-cells become exhaustive, unresponsive, and then sometimes recover [43]. Another explanation is that, at this timepoint, for the double-mutant vaccine, the circulating cytokine levels (cytokines quantified in the unstimulated samples) were at their highest, potentially masking any effect brought about by individual peptide stimulation pools. This explanation is favored for the double-mutant vaccine, as there were no significant changes in the T-cell surface marker differences at this timepoint (Figure 4). As compared to previous testing of single-deletion mutants, the double-deletion mutant seems to be more immunogenic, as it far out performed BCG in terms of response at the 8 WPV timepoint (the timepoint tested in both studies), although more robust side by side testing is needed to confirm this [20]. Finally, the CDC WT subcutaneous vaccination thoroughly underperformed in terms of immune response. It is well known that, even after becoming infected with *M. tuberculosis*, the natural immunity generated is not protective, although natural infection occurs via aerosol [44]. However, CDC WT was chosen as the backbone for this vaccine due to its immunogenic nature, despite being from a marginal lineage as compared to the epidemic-associated, BCG-resistant Bejing strains of *M. tuberculosis* [45,46]. *M. tuberculosis* CDC1551 has been shown to cause increased seroconversion without increased disease compared to other clinical isolates and, through a potent and early stimulation of Th1 responses in hosts, is thought to limit its own infection and have longer survival times despite having similar bacterial burdens to other strains [27,28,47]. Our single-gene-deletion LAV’s previously tested showed protection against a challenge of the *M. tuberculosis* Beijing 4619 strain, despite the different backbones [20]. Interestingly, our data show that vaccination with the CDC WT strain does not produce a significantly different response to BCG in terms of cytokine response.

We also examined how the LAVs modulated what types of T-cells developed over time. Unsurprisingly, given the cytokine data, the double-mutant vaccine produced higher Ag85B- and ESAT-6-specific T-cells than the controls and BCG where relevant. However, the Ag85B- and ESAT-6-responding T-cell populations characterized did not differ greatly between the groups, perhaps partially due to low levels of responding cells in the controls. However, there were distinct differences between the Ag85B and ESAT-6 populations within the groups. In particular, T-regulatory cells, which have negative associations with progressive TB [48], were more prominent in the Ag85B-responding cell populations. Resident memory cells (CD103^+^CD69^+^CD62^−^), which have previously been identified as key factors in increased protection from intranasal-BCG vaccination [49,50], were higher for 8 WPV and 32 WPV only in the ESAT-6 group. The 8 WPV timepoint for Ag85B also had increased levels of resident T-cells. Interestingly, when the resident T-cells were higher, so were the cytokine responses to the respective peptide stimulation. Overall, our surface cell data comparisons were hindered by small numbers of responding populations at some timepoints and noisy data. However, the trends observed here are still interesting for the in vivo dynamics brought on by subcutaneous vaccination.

Unsurprisingly, BCG was never detected in the lung, but did manage to persist in the spleen throughout the 32 WPV timepoint. BCG’s inability to reach the lung is not necessarily positive from an immunogenicity perspective, as this could impact the development of an immune response in the main environment for *M. tuberculosis.* Indicative of BCG’s inability to get to the lung, BCG-vaccinated murine lymphocytes were unreactive to Ag85B and TB10.4 stimulation. The stimulated samples remained at similar levels to the media stimulated levels on average across all timepoints and cytokines. At the later timepoints (16 WPV and 32 WPV), where TB10.4 stimulation was tested, BCG showed little TB10.4 reactivity. Of note, TB10.4 reactivity has been shown to be a correlate of immunity for BCG vaccination before in mice at earlier timepoints (such as 8 WPV) and, in other studies, has shown reactivity at even earlier post-vaccination timepoints [51,52]. Finally, BCG did not display any significant populations of polyfunctional T-cells, which have been identified numerous times as an important protective factor against a TB challenge [41,42].

Overall, we characterized the safety and immunogenicity of a live-attenuated vaccine. Although the safety of this particular vaccine construct was not established, the knockout structure and resulting data may be important for further research, given the double-mutant vaccine’s success in producing a robust immunological response, where even a WT strain administered subcutaneously could not. Further, the data generated from this study may also be important for vaccine development considerations in terms of how continuous antigenic stimulation with ESAT-6 and Ag85B led to certain T-cell populations. Finally, this study thoroughly characterized BCG vaccination in the murine model over an extended period of time, highlighting its continued safety, but lack of immunogenicity by the standards presented here.

## Figures and Tables

**Figure 1 microorganisms-11-02105-f001:**
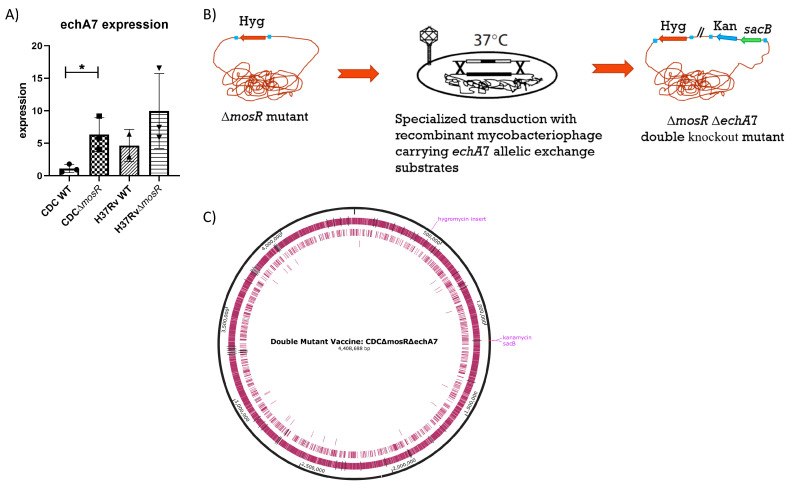
Relationship between *mosR* deletion and *echA7* expression. Cultures of *M. tuberculosis* CDCΔ*mosR* and *M. tuberculosis* CDC1551 WT and H37RvΔ*mosR* and H37Rv WT were grown to ~1OD before having their RNA extracted. qRT-PCR using gene specific primers was then used to identify *echA7* gene expression (**A**) normalized to an internal control and the average CDC1551 WT expression. A double gene deletion was generated in *M. tuberculosis* CDCΔ*mosR* using a specialized gene transduction process depicted in (**B**) to obtain *M. tuberculosis* CDCΔ*mosR*Δ*echA7*. (**C**) Map of double-mutant vaccine: hygromycin is inserted into the center of the *mosR* gene and kanamycin and *sacB* genes are inserted in place of the *echA7*. Image made with SnapGene^®^. Differences between expression levels were determined to be significant via unpaired *t*-test where a *p* value of <0.05 (*) is considered significant. The CDC WT is compared to CDCΔ*mosR* while H37Rv is compared to H37RvΔ*mosR*.

**Figure 2 microorganisms-11-02105-f002:**
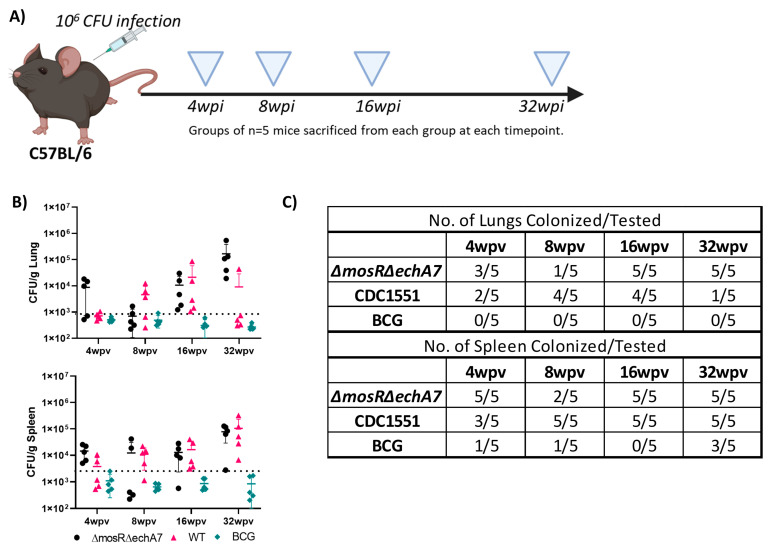
*M. tuberculosis*Δ*mosR*Δ*echA7* is able to persist in mice. (**A**) Figure depicting experimental design for testing immunogenicity and safety of subcutaneous infection with three mycobacterial strains. Mice vaccinated with 10^6^ CFU subcutaneously of *M. bovis* BCG (BCG), *M. tuberculosis*Δ*mosR*Δ*echA7* or *M. tuberculosis* CDC WT (CDC1551) were sacrificed at 4, 8, 16, and 32 weeks post-vaccination (WPV). Image made with Biorender. (**B**) Lungs and spleen from each group were harvested, weighed, homogenized, and plated on 7H10 for bacterial numeration, calculated as CFU (colony forming units) per gram. (**C**) As not all tissues collected had detectable levels and level of detection varied greatly among groups, number of tissues colonized are listed here.

**Figure 3 microorganisms-11-02105-f003:**
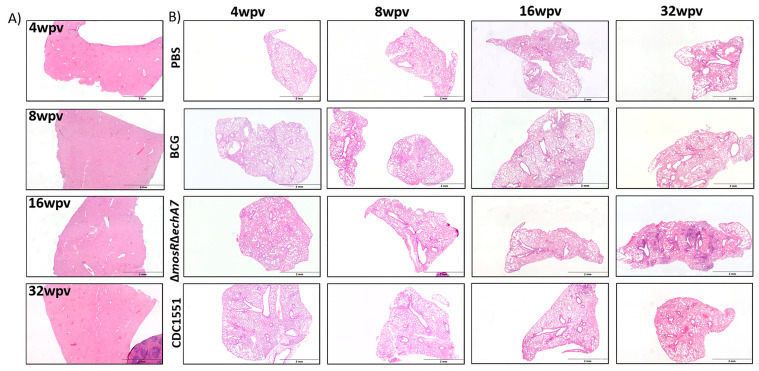
Histopathological outcomes of a double gene deletion *M. tuberculosis*. Mice vaccinated with 10^6^ CFU subcutaneously of *M. bovis* BCG (BCG), *M. tuberculosis*Δ*mosR*Δ*echA7* (double gene deletion) or *M. tuberculosis* CDC WT (CDC WT) were sacrificed at 4, 8, 16, and 32 weeks post-vaccination (WPV). (**A**) Livers were embedded, cut, and stained with H&E. Representative samples from each timepoint for the double-mutant vaccine at each timepoint shown. (**B**) Lungs from each group were embedded, cut, and stained with H&E. A representative sample from each timepoint and each group shown above.

**Figure 4 microorganisms-11-02105-f004:**
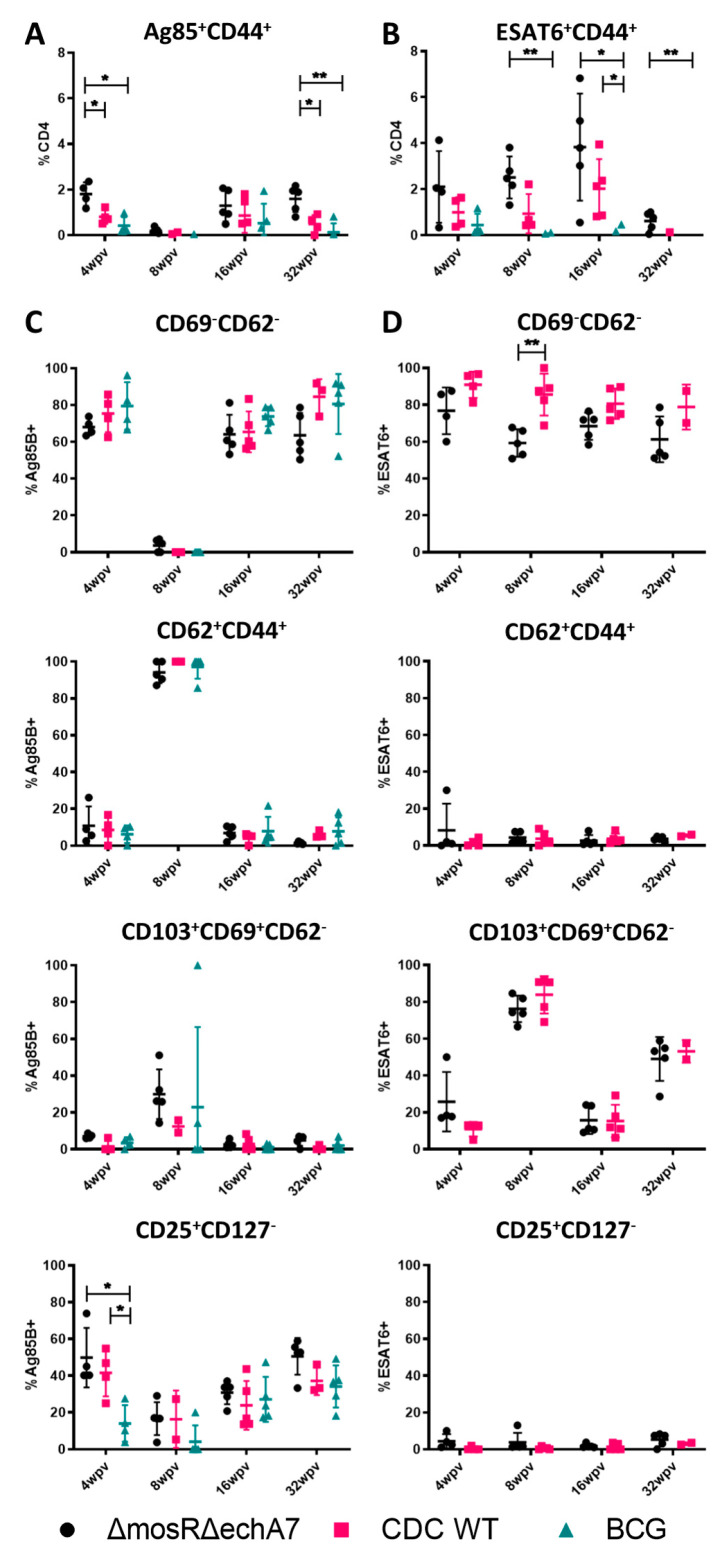
The immune profile of the double-mutant vaccine. Surface staining and flow cytometry analysis was performed at 4, 8, 16, and 32 WPV to assess T-cell populations. (**A**) Ag85B^+^ CD44^+^ CD4^+^ T-cells expressed as total percentage of CD4^+^ cells. (**B**) ESAT-6^+^ CD44^+^ CD4^+^ T-cells expressed as a total percentage of CD4^+^ T-cells. (**C**) Populations of effector (CD69^−^CD62^−^), central (CD62^+^), resident (CD103^+^, CD69^+^, CD62^−^), and regulatory T-cells (CD25^+^, CD127^−^) from the Ag85B-specific population. (**D**) Same as (**C**), except from the ESAT-6 specific population. Significance was determined via Tukey’s multiple comparisons test where *p* values of <0.05 (*) and <0.01 (**) are considered significant.

**Figure 5 microorganisms-11-02105-f005:**
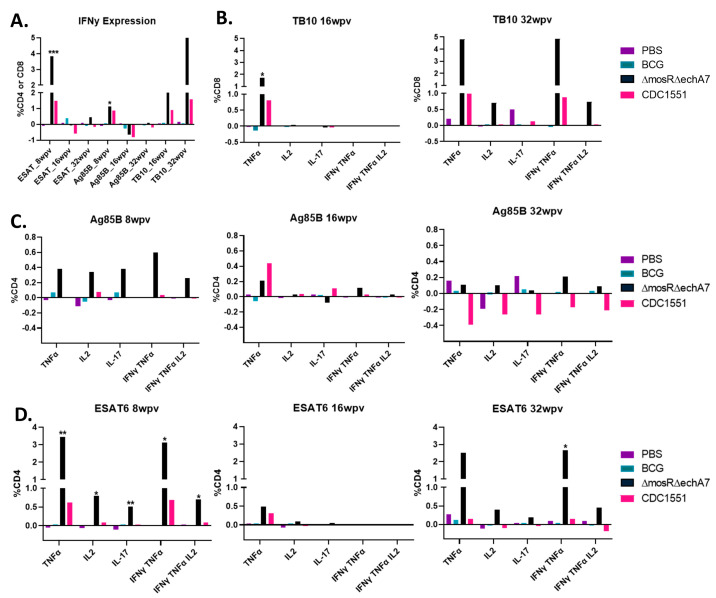
The double-mutant vaccine produces superior CD4^+^ and CD8^+^ T-cell cytokine responses. Intracellular cytokine staining was performed on lungs harvested at 4, 8, 16, and 32 WPI to assess T-cell responses post vaccination. Cells were stained for surface-markers CD4^+^ and CD8^+^ as well as intracellularly for IFNγ, TNFα, IL-2, and IL-17. All graphs show the difference between the stimulated and unstimulated levels for each timepoint and stimulant. (**A**) Depicts IFNy expression for all groups/timepoints as a function of %CD4 or CD8 cells in the case of TB.10. The other graphs depict the different cytokine levels as a percentage of CD8 T-cells for TB.10 (**B**) or as a percentage of CD4 T-cells for Ag85B (**C**) and ESAT-6 (**D**). Only significance as compared to PBS is marked above columns determined to be significant via the Kruskal–Wallis test. *p* values of <0.05 (*), <0.01 (**), and <0.001 (***) were considered significant.

**Figure 6 microorganisms-11-02105-f006:**
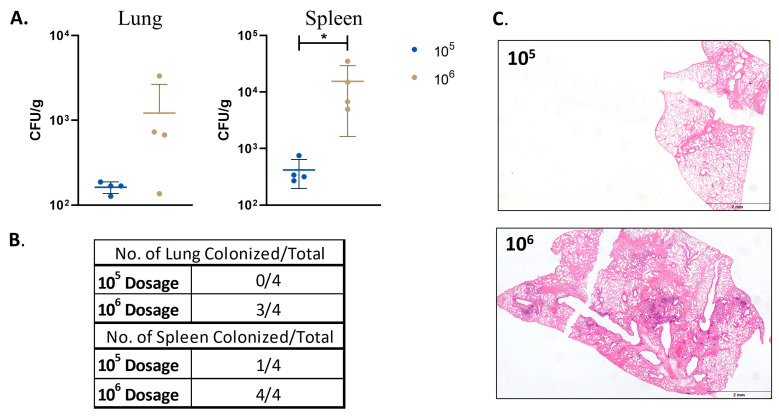
Safety and persistence of *M. tuberculosis*Δ*mosR*Δ*echA7* mutant in mice. Mice were infected with *M. tuberculosis*Δ*mosR*Δ*echA7* at either 10^6^ CFU (original dosage) or a 10^5^ dose. Mice were then sacrificed at 16 WPV and lungs and spleens were harvested for bacterial burden and histopathology. (**A**) Lung and spleen bacterial burdens. (**B**) Table of colonization percentages. (**C**) Representative lung images of embedded tissue cut and stained with H&E. Significance was determined via the Mann Whitney test where *p* values of <0.05 (*) were considered significant.

## Data Availability

All relevant data to this report are presented here or in the Appendix A.

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
