# Peer review of "The Immunogenicity and Safety of Mycobacterium tuberculosis-mosR-Based Double Deletion Strain in Mice"

_microorganisms, 2023, doi:10.3390/microorganisms11082105_

Round 1

Reviewer 1 Report

In their manuscript, Hildebrand and collaborators present the results of the evaluations of security and immunogenicity of the double-mutant Mtb  (Mtb CDC1551 ΔmosR ΔechA7). The manuscript is clear in their construction but some points need to be clarified. My major concern of this manuscript is that authors refer to the double-mutant Mtb as a live-attenuated vaccine, but the data they present suggest this is not an attenuated strain, because actually it causes a progressive infection similar to the parental strain. For their consideration, my commentaries below:

1. As written in lines 75-77, the only motivation of the authors to construct the double mutant is to comply the WHO standards based in the evidence of the efficacy of the separated mutants ΔmosR ΔechA7, but I think they were also looking to improve the efficacy of the vaccine. Please clarify this in the introduction.
2. The experimental design included the parental Mtb CDC1551, the double mutant, BCG as control, but they did not included groups of mice vaccinated with Mtb CDC1551 ΔmosR or Mtb CDC1551 ΔechA7. If authors  think the double mutation could provide higher benefit in terms of immunogenicity, they have to include these parallel groups to complete the description.
3. It is usual to do the experiments of safety of live-attenuated vaccines in immunocompromised mice. The use of immunocompetent mice can mask the effects on safety. More if we see in figure 2B that double mutant actually infects C57BL/6 mice. In this context, I suggest to authors do not guarantee the security of the double mutant because of 1) they made the evaluation in immunocompetent mice, 2) the mutant infect the lungs progressively and 3) it results in lung damage as shown in figures of panel 3B.
4. I suggest to authors to evaluate the replication of the double mutant in macrophages and compare it with the single-mutants and with BCG. This could give an idea of how replicant is the mutant before its use in animals.
5. The immunological evaluation is interesting, but we have to have in mind that the double mutant Mtb ΔmosR ΔechA7 actually is not an attenuated vaccine, it is a Mtb mutant which preserves its ability to infect lungs and cause tissue damage. Then, I suggest to authors re-orient their discussion to the effect of these deletions on the course of the infection with this double-mutant, but not in the context of vaccines, because the “vaccinated” mice are already in the course of an Mtb infection.

Author Response

Dear reviewer, Thank you for your thoughtful comments.  We've provided a comment by comment response below.

  1. As written in lines 75-77, the only motivation of the authors to construct the double mutant is to comply the WHO standards based in the evidence of the efficacy of the separated mutants ΔmosR ΔechA7, but I think they were also looking to improve the efficacy of the vaccine. Please clarify this in the introduction.
    1. Thank you for this thoughtful comment. We have added a paragraph to the introduction including references detailing these gene choices.  In brief, mosR was found to be an operon of genes used for late-stage survival for mycobacterium and highly upregulated at later timepoints in vivo.  EchA7 is part of a unique genomic island of 32 consecutive genes upregulated in vivo and is part of lipid metabolism for M. tuberculosis.  Both single KOs were attenuated when tested via aerosol infection and were protective when given subcutaneously to challenge with Beijing 4619, a high virulence clinical isolate.  These changes can be found on lines 61-77.
  2. The experimental design included the parental Mtb CDC1551, the double mutant, BCG as control, but they did not included groups of mice vaccinated with Mtb CDC1551 ΔmosR or Mtb CDC1551 ΔechA7. If authors think the double mutation could provide higher benefit in terms of immunogenicity, they have to include these parallel groups to complete the description.
    1. Thank you for this comment. In a prior study, we compared the single deletion mutants (CDC::mosR and H37Rv::echA7) to BCG at 8WPV timepoint.  This study is cited in the current manuscript (Marcus et al).  In this study, the single deletion mutants produced more IFNy than BCG but only significantly for the echA7.  The levels were much lower (~0.7% of CD4 T-cells for the H37Rv::echA7) than they were for the double deletion at 8WPC although the stimulant was whole cell lysate vs individual peptide pools.  Given these single deletion mutants had already been partially characterized, we chose CDC WT given subcutaneously as a “positive” control for immunogenicity given the lack of response we’d seen from BCG previously and knowing the single-deletions were only marginally better than BCG.  We have also added a line to our discussion noting the prior work and suggesting further side by side testing for confirmation (lines 407-411).
  3. It is usual to do the experiments of safety of live-attenuated vaccines in immunocompromised mice. The use of immunocompetent mice can mask the effects on safety. More if we see in figure 2B that double mutant actually infects C57BL/6 mice. In this context, I suggest to authors do not guarantee the security of the double mutant because of 1) they made the evaluation in immunocompetent mice, 2) the mutant infect the lungs progressively and 3) it results in lung damage as shown in figures of panel 3B.
    1. Thank you for this comment. We agree that immunocompromised mice are best for safety studies while immunocompetent are better for understanding the underlying immunological mechanisms.  In our study, we were more focused on the immune response and upon seeing the damage as noted by the reviewer in 3B, decided that this vaccine, at least in the current formula, was not safe enough to test in immunocompromised mice.  We note that “the safety of this particular vaccine construct is not established” in our discussion at line 455.
  4. I suggest to authors to evaluate the replication of the double mutant in macrophages and compare it with the single-mutants and with BCG. This could give an idea of how replicant is the mutant before its use in animals.
    1. Thank you for this comment. In future studies, we will likely test our double deletion mutants in macrophages to better understand why this construct has such low safety profile.  Yet, the presented work will be very beneficial to the tuberculosis vaccine discovery field.
  5. The immunological evaluation is interesting, but we have to have in mind that the double mutant Mtb ΔmosR ΔechA7 actually is not an attenuated vaccine, it is a Mtb mutant which preserves its ability to infect lungs and cause tissue damage. Then, I suggest to authors re-orient their discussion to the effect of these deletions on the course of the infection with this double-mutant, but not in the context of vaccines, because the “vaccinated” mice are already in the course of an Mtb infection.
    1. Thank you for this thoughtful comment. We chose to call the double deletion mutant here a “vaccine” due to: 1) it was developed as a vaccine and 2) it was administered via a route used for vaccination.  We also spent a lot of time discussing what to call it and settled on a live-attenuated vaccine candidate that ultimately was not attenuated at the dose given (though very attenuated at a lower dose).  Given the subcutaneous route, the immunological data is more relevant for vaccine studies as well.

Reviewer 2 Report

The main question addressed by the research is to explored the immune profile and safety of a live-attenuated M. tuberculosis construct with double deletions of mosR and echA7 genes. Mice were sacrificed to evaluate vaccine persistence, histopathology and immune responses, and this because the live attenuated vaccine M. bovis Bacillus Calmette-Guerin (BCG), the most administered vaccine worldwide, protects the children from disseminated tuberculosis, but it has limited efficacy in adults.

Developing effective vaccines against M. tuberculosis remains critical for tuberculosis control and a global health priority. Live-attenuated vaccines continue to be a strong area of potential development and this article show that CDCDmosR and H37RvDechA7 are promising vaccine candidates.

This study has characterized the safety and immunogenicity of a live-attenuated vaccine. Live-attenuated vaccine development balances safety with immunogenicity. Moreover, this study has demonstrated how just a small difference in dosing can affect the colonization results even if this study highlights both the need to evaluate dose as well as longevity of safety studies of live-attenuated vaccines.

I found the article very interesting, well done and publishable.
Kind regards.

Author Response

Dear Reviewer,

Thank you for your time and effort in critiquing our manuscript.  We appreciate your comments greatly!

Reviewer 3 Report

This study is within global reseach to develop new vaccines, including vaccines based on live attenuated strains. Following their previous research, the authors created a construct with two deletions of genes echA7 and mosR based on well-known strains CDC1551. The authors characterized safety and immunogenicity of this mutant strain in mice model using subcutaneous injection. Strain BCG and the wild-type (WT) parental strain M. tuberculosis CDC1551 were used references. In particular, they found that double mutant vaccine produced stringer IFNγ, TNFα, IL-17 and IL-2 CD4 and CD8 T-cell responses compared to BCG and CDC1551 WT strains.

The study is interesting, robust and relevant (although the result is rather negative - that particular vaccine was not suitable to further study, but the knockout structure and resulting data may be useful for further research since the double mutant vaccine produced a robust immunological response).

Comments:

(1) I would advice to provide a brief explanation and references why these two genes were selected as targets to be deleted: echA7 and mosR genes

(2) My general comment, or rather a suggestion, is to pay more attention to the more globally relevant strains when creating such live attentated vaccines. CDC1551 (as well as H37Rv) belong to phylogenetically marginal lineage. Instead, Beijing and LAM strains are frequently MDR and transmissible and globally spread. Development of vaccines based on such strains coming from high-burden countries, e.g. China and Russia (more generally speaking former Sovit Union) makes more sense, I think.

Author Response

Dear Reviewer,

Thank you for your time and effort in critiquing our manuscript.  We appreciate your comments greatly.  We have addressed your two main critiques below.

  1. Would provide a brief explanation and references to why these two genes were selected as targets to be deleted: echA7 and mosR genes

Thank you for this thoughtful comment.  We have added a paragraph to the introduction including references detailing these gene choices.  In brief, mosR was found to be an operon of genes used for late-stage survival for mycobacterium and highly upregulated at later timepoints in vivo.  EchA7 is part of a unique genomic island of 32 consecutive genes upregulated in vivo and is part of lipid metabolism for M. tuberculosis.  Both single KOs were highly attenuated when tested via aerosol infection and were protective when given subcutaneously to challenge with Beijing 4619, a high virulence clinical isolate.  These changes can be found on lines 61-77.

  1. Suggestion to pay more attention to relevant strains when creating live attenuated vaccines such as the Beijing strain.

Thank you for this suggestion.  We have added additional information into the discussion about the choice for using CDC1551 over other strains from lineages with higher disease burdens.  These changes can be found on lines 415-416, 420-422.  We also highlighted that in our prior evaluation of the single knockouts that they were shown to be protective against a challenge from Beijing 4619, a high virulence clinical isolate.